# Imaging of the Reconstructed Breast

**DOI:** 10.3390/diagnostics13203186

**Published:** 2023-10-12

**Authors:** Theodora Kanavou, Dimitrios P. Mastorakos, Panagiotis D. Mastorakos, Eleni C. Faliakou, Alexandra Athanasiou

**Affiliations:** 1Diagnostiki Mastou, 41222 Larissa, Greece; 22nd Breast Surgery Unit, Mitera Hospital, 15123 Athens, Greece; 3Athens Breast Clinic, 11527 Athens, Greece; 4Breast Imaging Department, Mitera Hospital, 15123 Athens, Greece

**Keywords:** breast, autologous reconstruction, implant-based breast reconstruction, MRI of reconstructed breast, complications of reconstruction surgery, breast cancer recurrence

## Abstract

The incidence of breast cancer and, therefore, the need for breast reconstruction are expected to increase. The many reconstructive options available and the changing aspects of the field make this a complex area of plastic surgery, requiring knowledge and expertise. Two major types of breast reconstruction can be distinguished: breast implants and autologous flaps. Both present advantages and disadvantages. Autologous fat grafting is also commonly used. MRI is the modality of choice for evaluating breast reconstruction. Knowledge of the type of reconstruction is preferable to provide the maximum amount of pertinent information and avoid false positives. Early complications include seroma, hematoma, and infection. Late complications depend on the type of reconstruction. Implant rupture and implant capsular contracture are frequently encountered. Depending on the implant type, specific MRI signs can be depicted. In the case of myocutaneous flap, fat necrosis, fibrosis, and vascular compromise represent the most common complications. Late cancer recurrence is much less common. Rarely reported late complications include breast-implant-associated large cell anaplastic lymphoma (BIA-ALCL) and, recently described and even rarer, breast-implant-associated squamous cell carcinoma (BIA-SCC). In this review article, the various types of breast reconstruction will be presented, with emphasis on pertinent imaging findings and complications.

## 1. Introduction

Nowadays, breast conserving surgery followed by radiotherapy is considered the standard of care for breast cancer treatment. Although the intensive mammographic screening has led to early identification of breast cancer, since the late 1990s, there has been an increase in mastectomy rate, accounting for 11.3% in the year 2020, 33% in women aged 31–40 years and 39.9% in women younger than 30 years old [1]. In recent decades, the implementation of magnetic resonance imaging (MRI) in the preoperative evaluation of breast cancer patients has outlined a more precise estimation of the extent of the disease that, on the one hand, has led to an 11% increase in mastectomy rate but, on the other hand, is counterbalanced by a 3% lower reoperation rate [2].

Mastectomy is not only performed for curative purposes but also prophylactically in women at high risk for developing breast cancer [3]. The rate of prophylactic mastectomy has increased to 35.7% for bilateral mastectomy and 22.9% for contralateral mastectomy [4].

Immediate breast reconstruction has positive effects on body image and psychosocial well-being and current guidelines recommend clinicians to offer immediate breast reconstruction to every patient with an indication for mastectomy, with similar oncological outcomes to delayed reconstruction [5,6,7]. Breast reconstruction should be available and proposed to all women (who are fit for this more major surgery) requiring mastectomy. There is a high degree of satisfaction and psychosocial well-being in the long term compared with mastectomy and no immediate breast reconstruction [8]. During recent years, numerous methods have been employed for breast reconstruction, tailored to each patient’s needs and aiming for a lower complication rate with improved aesthetic outcomes. The many reconstructive options available and the changing aspects of the field make this a complex area of plastic surgery, requiring knowledge and expertise in many different reconstructive options, but also familiarity with the technical improvements influencing patient care.

Ideally, one should strive for a treatment roadmap, even before mastectomy. An algorithm should be developed to include any eventuality and enable the discussion of options with the patient. This process starts with the evaluation of the patient’s health and body, assessing donor sites, the availability of fat deposits, the scars, and possible radiation treatment of the other breast. The patient’s age and family history will also be very crucial. Comorbidities, such as uncontrolled diabetes mellitus, obesity, cardiac disease, hypercoagulation, and smoking, are related to increased rates of complications that could compromise autologous reconstructive surgery, whereas stage T4 tumors invading the chest wall have a negative profile for breast reconstruction. The possibility of hereditary breast cancer needs to be considered when facing a significant family history or a young age at diagnosis [9,10].

Issues to be discussed with the breast caring team and the patient should include the type of surgery performed, the type of incision to be used, whether one should opt for a delayed or an immediate reconstruction, as well as the timing of each intervention. At the same time, the possibility of a bilateral surgery, either reconstructive or even prophylactic, needs to be taken into account and may steer the reconstructive team towards choosing options that will be able to accomplish that [11].

Autologous tissue reconstruction is preferred in patients undergoing postmastectomy radiotherapy, due to the lower risk of complications compared to implant reconstruction. Since postsurgical radiotherapy is not an absolute contraindication for implant-based reconstruction, these patients opt for the placement of a tissue expander only, whether in the pre- or retro-pectoral position. In this case, the patient has the experience of the prosthetic reconstruction and may decide to proceed, or switch to autologous reconstruction after all oncological treatment is delivered, or if prosthetic reconstruction fails. At the same time, all treatment is completed with the expander in place, sparing the potential autologous tissue from being exposed to radiation therapy [12,13].

The question of immediate or delayed reconstruction is also very significant. There are very few absolute contraindications for immediate breast reconstruction, such as inflammatory breast cancer. Even in the setting of metastatic disease, one should discuss this with the patient. A possible reason to avoid breast reconstruction is the possibility that a complication might delay the delivery of chemotherapy or radiation treatment.

Care should be taken to address the expectations of the woman and try to inform her regarding limitations and shortcomings of each method of reconstruction. The issues regarding breast implants, including breast-implant-associated atypical large cell lymphoma and squamous cell carcinoma (BIA-ALCL and BIA-SCC), should be presented in their true proportions to answer patients’ concerns, including what should be done to thwart such an outcome.

The breast reconstruction options are summarized in three major categories: autologous reconstruction; alloplastic reconstruction; and pure fat transfer reconstruction. Autologous reconstruction and alloplastic reconstruction could be complemented by fat transfer or by the use of meshes: resorbable or non-resorbable; biologic or synthetic.

It is important to understand the different surgical techniques for breast reconstruction in order to be familiarized with the normal postoperative imaging appearance of the reconstructed breast and also to recognize common benign complications associated with each reconstruction method. The identification of breast cancer recurrence and malignant disease associated with breast implants is another issue that is discussed in this review.

## 2. Surgical Techniques for Breast Reconstruction

### 2.1. Autologous Reconstruction

Although autologous breast reconstruction is considered the best available option (since the neobreast has a more natural ptotic appearance, feel, and texture), about 80% of reconstructions are performed differently. Autologous reconstruction is most often used in the case of previous exposure to radiation treatment. The poor vascularity and elasticity of tissues make the use of expanders and implants more difficult, which is reflected in the higher percentages of failure in such scenarios. The added vascularity of the autologous tissues provides a significant improvement in the healing potential of such tissues, leading to uneventful healing and successful reconstructions that will tolerate the test of time. Another scenario where autologous tissues present their advantages is that of delayed reconstruction. In this case, especially after a horizontal scar mastectomy with removal of all excess skin, the deficit of coverage is the most significant factor for a successful reconstruction. Depending on tissue elasticity, one could consider alloplastic reconstruction, but in the setting of significant radiation changes and extensive scarring or tight tissues, the ample tissue provided by autologous reconstruction excels. Unilateral reconstruction is also an area where autologous reconstruction offers significant advantages. However, one should keep in mind that this requires adequate donor areas and that there will be some amount of transplanted skin that will be visible, creating a patchwork appearance on the breast, of a quality difficult to predict.

#### 2.1.1. Transverse Abdominal Myocutaneous Pedicle (TRAM) Flap

The donor site is the lower abdomen and TRAM flaps are harvested with two techniques, either as a pedicle flap with blood supply from the superior epigastric vascular system, or as a free flap based on the inferior epigastric vasculature (Figure 1). The TRAM flaps in both cases contain fatty tissue, muscle, fascia, and vessels. The pedicled TRAM flap is rotated on its vascular pedicle from the lower abdomen to the mastectomy site, whereas the free flap is completely separated from its abdominal blood supply and positioned at the mastectomy site where it is anastomosed to thoracodorsal or internal mammary vessels [14].

#### 2.1.2. Latissimus Dorsi Myocutaneous (LD) Flap

The latissimus dorsi is harvested from the middle back and, after identification of the thoracodorsal vessels, the myocutaneous or myofascial flap is transferred to the mastectomy site though a subcutaneous tunnel from the axilla [15]. Most often, the combined use of an implant is required to augment its size, or, more recently, fat grafting is used to achieve the same goal [16] (Figure 2). The LD flap has slowly fallen out of favor, especially as a primary reconstruction method. It is still favored as a salvage solution, particularly in radiation induced complications after reconstruction, and as a first line method for patients that are not eligible for TRAM flap reconstruction due to prior abdominoplasty, slim patients with insufficient abdominal tissue, and patients with comorbidities such as diabetes, obesity, or tobacco use.

More innovative surgical techniques, such as the extended LD flap [17], the scarless approach [18], and the muscle sparing LD flap [19], provide excellent aesthetic outcomes with lower flap-related complications.

#### 2.1.3. Muscle Sparing Free Flaps

With the workhorse TRAM flap, reconstructive surgery has moved on to other options; namely, the muscle sparing flaps, where a large part of the rectus muscle is preserved (free TRAM flap), and then the free alternatives of the same flaps, such as the deep inferior epigastric perforator (DIEP) flap, where the skin and fat are harvested together with their vessels but with no muscle [9]. At the superficial inferior epigastric artery (SIEA) flap skin and subcutaneous tissue are excised from the lower abdomen, with vascular supply from the superficial subdermal vascular plexus arising from the superficial inferior epigastric vessels, while the rectus muscle and fat are spared [20]. These flaps will require the use of microsurgery and prolonged operation time but because the muscle is spared there is reduced donor site morbidity and better patient recovery [20,21,22].

Many more flaps have been described, in case the abdominal tissues are not available (as after an abdominoplasty) or not sufficient. These include upper thigh flaps (transverse upper gracilis—TUG flap), gluteal flaps (gluteal artery perforator—GAP flap), and even lumbar tissue flaps [23]. All these options are free flaps, requiring microsurgery and careful dissection during harvesting. Another option, if a little more exotic, in case of tissue paucity, is the use of more than one flap per breast, i.e., the stacked flaps option. In this scenario, one would combine two flaps to achieve the reconstruction of one breast of larger size [24].

### 2.2. Alloplastic Reconstruction

#### 2.2.1. Implants

Implants containing saline or silicone have been the mainstay of breast reconstruction in recent decades. With the evolution of the 3rd and 4th generations of silicone gel implants, the use of saline implants is limited, accounting for only 4% of cases worldwide in 2016 [25]. There is a wide variance in silicone implant filler, shell, texture, and shape, and development is ongoing in the pursuit of an optimal device aimed towards the limitation of complications and an improved aesthetic outcome. The 6th generation of implants was introduced in 2011 (such as Motiva implants, Establishment Labs), having a smooth and uniform surface design that reduces chronic inflammatory foreign body response [26]. Silicone implants may consist of a single lumen or a double lumen with a silicone-filled inner chamber surrounded by a saline filled outer chamber that can be expanded through a valve [27].

Almost any postmastectomy patient is a candidate for a prosthetic implant reconstruction, but there are limitations in patients that will receive radiation therapy or have already been irradiated, since the irradiated breast is less elastic and there is an increased risk of complications [13].

Breast reconstruction can be performed in a single stage or in two stages, and in either case immediately after mastectomy or delayed. The double-stage reconstruction is recommended for women with small breasts or insufficient postoperative skin flaps. An implant with an expander is initially inserted, which gradually inflates to create a pocket and is replaced with a permanent implant during a second operation [28]. In addition to the well-known breast tissue expanders with an incorporated magnet in their filling port, making them incompatible with MRI, there are now tissue expanders available with non-magnetic ports, incorporated in their anterior wall, that are compatible with MRI [29]. There are also temporary expanders with remote ports, which are usually compatible with MRI and double-lumen permanent tissue expanders, with remote filling ports that possess an outer silicone gel layer and an inner expansile cavity, connected to the filling port [30].

Since the advent of BIA-ALCL cases and their clustering in patients with aggressively-textured surface devices, a classification of the types of breast implant surfaces has emerged, separating them into macro-, micro-, nano- and smooth-surface implants. Based on surface area characteristics and measurements, the textures are separated into: smooth/nano-texture (80–100 mm^2^); micro-texture (100–200 mm^2^); macro-texture (200–300 mm^2^); and macro-texture-plus (>300 mm^2^). Most, if not all, cases have been diagnosed in women exposed to macro- (mostly) or micro-textures. Nano and smooth are very rarely, if at all, present. This is another factor that has been impacting the choice of implants when considering breast reconstruction patients.

#### 2.2.2. Acellular Dermal Matrices (ADMs)

Since the development of ADMs, grafts prepared from human or pig skin or bovine pericardium pretreated to remove the cells, the options for breast reconstruction have multiplied. For a little more than a decade, traditional reconstruction with the implants placed mostly or totally under muscle coverage has been challenged [31]. To begin with, the placement has moved to the dual plane position, meaning it is still under the cover of the pectoralis major muscle cephalad, but now the lower pole is under the coverage of the mesh, most often an ADM, an allo- or xeno-graft (Figure 3).

At the same time, synthetic meshes appeared, assuming the same role as above. These could be resorbable or not, but they were man-made except for one which was composed of silk fibers. The benefits of such an approach include the decreased pain and tightness in the lower pole of the reconstructed breast, while at the same time there is a more natural shape to the inframammary fold. In the case of ADMs, there has been evidence of better tolerance of radiation treatments as well as less capsular contracture than without them [32]. Once more experience was accumulated with ADMs, the next step was to combine many sheets and prepare a circumferential “ravioli” type coverage of the expander or implant, and place them in the subcutaneous or prepectoral level (Figure 4). Despite all their noteworthy characteristics, which may actually turn out to include protection against BIA-ALCL, one of the obvious drawbacks of ADMs is their cost and the higher chance of the patient developing a seroma or infection.

### 2.3. Autologous Fat Reconstruction

Autologous fat grafting, also referred to as lipofilling or lipotransfer, is a novel approach in breast reconstruction, alone or usually in combination with breast implants to improve breast contour and volume. Other indications include the treatment of postmastectomy pain syndrome, capsular contracture pain, and post-irradiation fibrosis [33]. Fat is harvested with liposuction, usually from the flanks and abdomen. In the USA, over 30,000 cases of autologous fat grafting were reported in 2018. Common mammographic findings are fat necrosis (0–50%), calcifications (0–45%), and scar (1.5–28.5%) [34].

### 2.4. Nipple Areola Complex (NAC) Reconstruction

Many attempts are made to preserve the NAC when certain criteria apply. It seems that this attempt at preservation is employed more often with tumors closer to the NAC, provided imaging and pathology confirm the negative margins, with good results. In the cases where preservation seem unwise, one proceeds with NAC reconstruction. The simplest approach is 3-D tattooing of the NAC, which provides a good or even excellent visual effect. On MRI, the tattoo can produce a blooming artifact with mild heterogenous enhancement [35].

## 3. Preoperative Imaging

To prepare for an autologous tissue reconstruction, all members of the treating team should be thoroughly familiar with the procedure. In particular, the use of a perforator flap and preoperative imaging, although probably not always required, will certainly speed up the procedure by allowing the surgical team to focus on the best candidate vessels rather than explore all possibilities. CT and MR scans provide exceptional images with detailed coordinate location of the vessels, allowing dissection to proceed in a speedy fashion [36,37]. CT angiography and MR angiography can create a preoperative vascular roadmap outlining potential anatomical variances that will optimize the decision making and flap excision [38,39,40]. Color Doppler and/or Duplex ultrasound (US) will provide some help in the absence of the above modalities [41,42]. Recent advances in preoperative imaging report that laser-assisted indocyanine green fluorescence angiography (LA-ICGFA) and dynamic infrared thermography (DIRT) can successfully identify the dominant vessels in autologous reconstruction [38,43].

The use of clinical pathways can assist delivery of excellent patient care in all complex surgical procedures, and this is a case in point. Steps to minimize patient discomfort such as local anesthetic infusion pumps, blocks, or use of long-acting local anesthetics can go a long way towards prompting the mobilization and discharge of the patient.

## 4. Normal Imaging Appearance of the Reconstructed Breast

### 4.1. Autologous Reconstruction

On mammography, the reconstructed breast consists predominantly of fat and, in case of muscle-baring flaps, muscle strands can be visible. Postoperative scarring and clips are common findings (Figure 5) [44].

The high-resolution MRI images allow excellent imaging of the reconstructed breast, providing anatomical details that can differentiate the various surgical techniques used for autologous reconstruction. The MRI protocol should include unenhanced fat-saturated T1-weighted sequences, non-fat-saturated T2-weighted sequences, dynamic contrast enhanced fat-saturated T1-weighted sequences, and delayed sagittal fat-saturated T1-weighted sequences. Homogeneous fat suppression is of paramount importance, and should be considered a prerequisite for MR imaging of the reconstructed breast.

The pedicled or free TRAM flap and the LD flap consist of the rectus abdominis and latissimus dorsi muscle, respectively, and also the overlying skin and subcutaneous fat. In the TRAM reconstruction, the flap is recognized centrally along the anterior chest wall in axial images (Figure 6). In the LD reconstruction, the flap is more eccentric, with a tailed appearance of the muscle in the lateral breast as a result of the flipping and tunneling of the flap harvested from the back, differentiating it from a TRAM flap (Figure 7) [35]. In both cases, the muscle atrophies over time. A thin, low signal intensity, curvilinear line parallel to the breast contour is often visible, best appreciated on sagittal images, representing the dermal layer of the lower abdominal or the dorsal wall [35,45]. The contact zone of the TRAM flap to the mastectomy site, that corresponds to the musculovascular pedicle, may exhibit contrast enhancement [46].

At the muscle sparing free reconstruction flaps (DIEP, SIEA, GAP, TUG), the breast is replaced by fatty tissue from the lower abdomen, gluteal region, and thighs and, on MRI, the reconstructed breast consists of fat and a thin vascular pedicle anastomosed with the internal mammary artery [35]. The absence of muscular components can differentiate them from TRAM and LD flaps.

### 4.2. Implant Reconstruction

First and foremost, it is important to have knowledge of the type of implant that has been used for breast reconstruction and to identify the different normal imaging findings of each implant type.

A common area of confusion is the presence of a double-lumen implant (Mentor Becker implants, Allergan Style 150) compared to a single-lumen implant, or the use of a tissue expander, especially with a remote filling port. These implants are filled partially (double-lumen) or totally (single-lumen implants, tissue expanders) with normal saline. They will (particularly the double-lumen ones) sometimes be confused with a possible rupture, especially if the type of implant is unknown. The possible double-lumen capsule, common especially in the macro-textured surface implants (as described above, such as Allergan Biocell or Mentor Siltex, which are unfortunately associated with BIA-ALCL as well) can sometimes be confused with a possible implant leak or rupture, leading to implant exchanges. Some of these implants are no longer available but could still be encountered by radiologists in their daily practice. Newer generation implants, namely the 6th generation implants such as Motiva (Establishment Labs), produce a very minimal capsular tissue response, affecting the radiological picture, while at the same time incorporating numerous imaging characteristics, assisting in the diagnosis of implant rotation. B-Lite implants (Polytech) allow for a clearer mammogram since they contain a microsphere technology which is more radiolucent.

A common question is whether a tissue expander or an expander-implant are compatible with MRI. As a rule, the remote port expanders as well as the expander-implants feature MRI compatible ports, as they are located by palpation. On the other hand, ports incorporated in dedicated breast tissue expanders usually contain magnetic locating elements, rendering them MRI non-compatible. A notable exception is the Motiva Flora^®^ tissue expander, which has an integrated radio frequency identification (RFID) port with radio-wave identification technology, allowing it to be exposed to the MR field without difficulty.

The typical mammographic appearance of an implant is a radiopaque oval mass with smooth margins whose density varies depending on the filling material (Figure 8a). A band of soft-density tissue surrounding the implant, with or without calcifications, represents the thick fibrous capsule that is formulated after the implant insertion as a result of a foreign body reaction. Folds within the implant and the valve may be visible with the appropriate mammographic technique [47].

Ultrasonographically, both saline and silicone implants are anechoic, and the shell appears either as one echogenic line or as parallel echogenic lines. Internal folds may be recognized as wavy lines without disruption. The fibrous capsule is visible as an echogenic line parallel to the implant’s shell, sometimes with calcifications producing focal acoustic shadowing (Figure 8b). A small peri-implant fluid effusion is a normal finding. In implants with expanders, the valve is visible and caution should be taken so that partially expanded implants should not be mistaken for ruptured implants [47].

MRI has a high spatial resolution and is the most accurate modality for evaluation of implants. Another advantage of MRI over conventional imaging modalities is the ability to enhance or suppress the signals of water, silicone, and fat. The most frequently used MR sequences are a fast T1-weighted multiplanar sequence, a T2-weighted fast spin-echo sequence, silicone-only sequences (silicone high signal intensity, water low signal intensity) and silicone-saturated sequences (silicone low signal intensity, water high signal intensity) [48,49]. A proposed MRI protocol is presented in Table 1. The contrast administration and dynamic sequences are not obligatory for implant imaging and can be performed only in case of associated findings.

A single-lumen implant has an intact shell and is surrounded by a thin fibrous capsule with low signal intensity on all sequences. Saline implants appear with water signal intensity on all sequences, and a valve is recognized within the lumen with low signal intensity, while silicone implants show high signal intensity on T2-weighted and silicone-selective sequences and low signal intensity on T1-weighted sequences (Figure 9). A double-lumen implant appears with an inner chamber of silicone with high signal intensity and an outer chamber of saline with water signal intensity (Figure 10) [47,49].

Internal radial folds and a small amount of peri-implant reactive fluid are normal findings [27,48]. Radial folds are infoldings of the shell extending from the periphery of the implant, usually as a result of implant contracture, and are characterized as simple when they are short and straight, and complex when they are longer and curved, which may mimic an intracapsular rupture (Figure 11). Radial folds have a sheetlike appearance and a more perpendicular orientation to the imaging axis, while a rupture is more parallel. It is important to review multiple sequential images on all planes in order to recognize the folds communicating with the shell and differentiate them from intracapsular rupture [50].

## 5. Complications

### 5.1. Autologous Reconstruction

The complications in autologous reconstruction are divided into two categories: complications of the flap and complications of the donor site. The most common benign complications of autologous flaps are edema, seroma, hematoma, skin thickening, fibrosis, and fat necrosis.

#### 5.1.1. Seromas and Hematomas

Seromas and hematomas are common early postoperative complications that usually resolve gradually, although they may persist for months or even years after surgery [45]. Seromas are typical multilocular fluid collections with a high T2 signal on MRI, while hematomas present with variable appearances depending on the age of the blood products.

#### 5.1.2. Skin Thickening and Fibrosis

Skin thickening is usually present within 6 months after radiation therapy, as a result of impaired venous and lymphatic drainage, and resolves after 2 to 3 years [51]. On MRI, the thickened skin appears as a band with a low T1 signal and high T2 signal that is uniform and not very intense [45].

Fibrosis occurs gradually within 1 year after radiation therapy. The differentiation of irregular fibrotic masses from recurrent breast cancer is challenging and, although they share common imaging characteristics on mammography, MRI can be helpful. Fibrosis has a low signal intensity without enhancement or minimal gradual enhancement, whereas a recurrent tumor is isointense or slightly hyperintense, exhibiting strong contrast enhancement with washout kinetics [35,45]. One should keep in mind, however, that granulation of tissue can increase for up to 1 year or even longer after surgery, causing diagnostic dilemmas and the need for histological confirmation after biopsy [52].

#### 5.1.3. Fat Necrosis

The initially dominant reconstructive technique, the pedicle TRAM flap, was associated with increased donor site morbidity, leading to adoption of less invasive techniques without muscle excision like the free TRAM flap and the DIEP flap. However, the preservation of the rectus abdominis muscle produces flaps with fewer perforator vessels, thus compromising the flap vascularity and increasing the risk of perfusion-related complications. Fat necrosis is a result of ischemia due to insufficient arterial flow and poor venous drainage, with an incidence reaching up to 35% [45,51,53]. Free TRAM flaps and DIEP have a more robust vascular supply than pedicled TRAM, but anastomotic thrombosis has been reported at a rate of 2.4–6.3% in microsurgery, which can lead to the development of fat necrosis [54]. A recent meta-analysis of flap perfusion of 1891 pedicled versus 866 free TRAM and 1211 DIEP flaps showed that the free TRAM flap demonstrated lower risk of fat necrosis than pedicled TRAM, although there was no difference between DIEP and pedicled flaps [21]. Flaps with increased weight have a greater risk for fat necrosis, with odds increasing by 1.5 times for every 100 gr increase in flap weight, that can be offset with an increased number of perforators [55]. Surgeons should be cautious in technique selection, taking into account the patient’s body mass index, comorbidities, and tobacco use, which can contribute to the development of fat necrosis.

Fat necrosis exhibits a wide spectrum of imaging appearances evolving over time, that in some cases mimic local recurrence. In mammography, a well circumscribed radiolucent mass is the most frequent finding, but fat necrosis can present with suspicious pleiomorphic microcalcifications accompanied by a mass with irregular or spiculated margins due to pronounced fibrosis [56]. Ultrasonographically, it can appear as a cyst, a complex cystic lesion (Figure 12a), or a mass with indistinct margins (Figure 13a). The absence of vasculature in Color Doppler helps in the differentiation of fat necrosis from tumor recurrence [57].

The most common appearance of fat necrosis on MRI is an oval mass isointense to fat in all sequences, with a low T1 signal on fat-saturated images, usually non-enhancing (Figure 12). The most challenging feature is the enhancement of granulated tissue with focal or irregular configuration. The degree of enhancement is variable, depending on the severity of inflammation, and kinetic analysis may be confusing (Figure 13). Rapid initial uptake and even washout of the contrast media have been reported, similar to malignancy [56,58]. A reliable feature that differentiates rim-enhancing fat necrosis from necrotic cancer is that the central non-enhancing area of cancer has a high T2 signal, while the central area of fat necrosis is fat that can be confirmed from the review of the non-contrast images. Fat has an equal signal intensity to fat in non-fat suppressed T1 images and a characteristically low signal in the STIR sequence, a fast spin echo sequence with inversion recovery that allows fat suppression (Figure 13). This is characterized as “black hole” sign and is pathognomonic of fat necrosis [59].

#### 5.1.4. Donor Site Complications

The incidence of donor site complications ranges from 7.7% to 38% for pedicled TRAM flaps and 17.9–24.7% for free TRAM flaps [60]. The muscle excision at the pedicled TRAM flap contributes to abdominal wall weakness, leading to abdominal wall hernia in 16% of patients [51]. At the muscle sparing flaps, although the muscle is preserved and the abdominal wall weakness is minimized, abdominal budge is nevertheless developed in 11.25% of free TRAM flaps and 8.07% of DIEP [61]. Factors related to abdominal wall bulging include patient age, comorbidities, previous abdominal surgery, operative time, and chemotherapy. A BMI greater than 23 kg/m^2^, and a US-measured thickness ratio of rectus abdominis muscle, evaluated at exercise, between the donor and normal site of less than 49% are reported as factors related to asymptomatic exercised abdominal wall bulging [62].

### 5.2. Implants and Alloplastic Reconstruction

Complications may manifest early or may be delayed after implant reconstruction. Seromas, hematomas, and infections are the most common early complications. The late complications include capsule contracture, bulging and herniation, implant rupture, gel bleed, fat necrosis, and malignant conditions such as BIA-ALCL and others. 

A diagnostic checklist for breast MRI after implant reconstruction is presented in Table 2, that will guide radiologists in identifying the type of implant, normal imaging findings and implant related complications.

#### 5.2.1. Seromas

Seromas are the most common early postoperative complications and, although they are usually absorbed within 4–5 months after surgery, they can persist for up to 1 year after. Seromas are recognized on US as peri-implant fluid collections with high signal intensity on T2-weighted images and low signal intensity on T1-weighted images on MRI. Although ADMs have been suspected to cause seroma formation, such a connection has not been established [63].

#### 5.2.2. Hematomas

The incidence of hematomas after breast augmentation ranges from 0.2 to 5.7% [64]. The appearance of hematomas on MRI is variable depending on their age, with acute and subacute hematomas appearing with high signal intensity on T1-weighted images without contrast enhancement [48].

#### 5.2.3. Infection

Infection is a significant complication reported in 5.8–28% of implant reconstructions, which may contribute to reconstruction failure [65,66]. Acute infections occur immediately after surgery and have the clinical signs of cellulitis [67], while subacute and late infections occur months to years after surgery and are usually clinically occult [68]. An increased infection rate is reported with the auxilliary use of ADMs that is even higher in respect to ADM burden [69]. Infection is usually manifested with a peri-implant effusion, while on MRI the capsule may present thickening and contrast enhancement (Figure 14).

#### 5.2.4. Capsular Contracture

Capsular contracture is one of the most common complications, occurring in 5–19% of patients after reconstruction [70]. After placement of the implant, a band of fibrous tissue is formulated as a result of a foreign body reaction. Capsular contracture occurs when the fibrous capsule becomes excessive and irregular, which causes abnormal contraction [71]. The 3rd and 4th generations of implants, and implants with smooth surfaces were more prone to capsular contracture [72]. A higher incidence is reported for implants with retro-pectoral positioning [73]. Radiation therapy may also induce capsular contracture [74]. The diagnosis is mostly clinical. Imaging findings suggestive of capsular contracture are a thickened irregular fibrous capsule, occasionally enhancing on MRI, spherical shape of the implant, and increased radial folds [49]. Clinical examination remains the gold standard for the estimation of capsular contracture and the Baker classification provides a standardized scoring system:Grade I—normal soft tissues implant texture;Grade II—firm texture with normal contour;Grade III—firm texture with altered contour;Grade IV—firm texture with altered contour with concomitant pain [75].

#### 5.2.5. Implant Rupture

Implant rupture is a well-known potential complication occurring in both saline and silicone implants. The rupture of saline implants is clinically dramatic since the implant loses volume and the breast is deformed, and it is best described by the term “deflate” rather than rupture. Because the saline is absorbed from the body, the absence of imaging findings is quite frequent [76]. On the other hand, silicone implant rupture is often asymptomatic, especially intracapsular rupture that is evident only on imaging. The incidence of implant rupture is related with the age of the implant with a 12-fold increased prevalence for rupture of implants between 16 and 20 years of age, compared with implants between 3 and 5 years of age [77]. A minimum of 15% of modern implants are expected to rupture from 3 to 10 years after implantation and for implants intact 3 years after implantation, there is an estimated rupture-free survival of 98% at 5 years and from 83% to 85% at 10 years. Double-lumen implants have a lower prevalence of rupture compared with single-lumen implants [78].

There are two types of silicone implant rupture: the intracapsular rupture, where the implant’s shell is ruptured and silicone is leaking outside the shell but remains within the intact fibrous capsule (up to 78% of reported ruptures); and the extracapsular rupture, where silicone is leaking from a breach in the fibrous capsule outside of the implant to the surrounding breast tissue (up to 22% of reported ruptures) [79].

Mammography is the least sensitive method for detection of implant rupture with a reported sensitivity of 11–70% [80,81]. Mammography is unable to depict an intracapsular rupture due to the lack of visualization of the internal structure of the implant. A subtle sign of extracapsular rupture is the deformity of the implant, while radiopaque silicone inside the breast can be easily detected making a definite diagnosis of extracapsular rupture with an increased specificity of up to 89% [82].

Ultrasound is a widely available, cost-effective method for implant evaluation [83], with a reported sensitivity of 30–75% [80,84,85]. The “stepladder” sign is the most reliable sign of intracapsular rupture, represented by a series of echogenic lines coursing parallel to the probe in the anterior of the implant, produced by the ruptured shell [80,86]. Other signs such as the “keyhole” sign and the “subcapsular line” sign are early signs of intracapsular rupture but the differentiation from radial folds is challenging [79]. The “snowstorm” artifact, recognized outside the implant, is a sign of free silicone within the breast after extracapsular rupture. A snowstorm appearance can also be encountered at the axillary lymph nodes (Figure 15). The presence of silicone at the lymph nodes is not by itself solid evidence of extracapsular implant rupture since it may be the result of “gel bleed”. “Gel bleed” is actually a misnomer that describes the leakage of silicone at a microscopic level though a weakened but intact polymer shell [27,86,87].

MRI is considered the gold standard for the evaluation of silicone implant integrity, with a perfect sensitivity of nearly 100% but with lower specificity of 63% to 97% [48,49,81,82]. Several signs of intracapsular rupture have been reported and the “linguine” sign is most reliable with a sensitivity of 96% and a specificity of 94% [88]. The collapsed elastomer shell of the implant floats inside the silicone and it is depicted by curvy lines of low signal intensity within the silicone (Figure 16). Other definitive signs of intracapsular rupture are the subcapsular lines, low signal intensity lines parallel to the fibrous capsule surrounded by silicone. The “keyhole or teardrop” sign, the “salad-oil” sign and the “rat-tail” sign are possible signs of intracapsular rupture. The “keyhole and teardrop” signs are focal silicone invaginations between the implant shell and the fibrous capsule caused by a focal tear of the shell (Figure 9c). Although they are considered an early sign of uncollapsed intracapsular rupture, they are non-specific [27,47,49,79,89]. The “salad-oil” or “droplet” sign describes the silicone gel mixing with droplets of peri-implant fluid (Figure 9d). Contour irregularities and deformities such as the “rat-tail” sign of silicone extending along the chest wall are also non-specific signs of rupture [81].

Extracapsular rupture involves the rupture of both the implant shell and the fibrous capsule, with leakage of silicone to the surrounding tissues [27,79]. The free silicone is best recognized in the silicone-selective MR sequences as areas of high signal intensity outside the fibrous capsule (Figure 17a). Silicone may also migrate through the lymphatics to the axillary lymph nodes that can be perceived in the silicone-selective sequences (Figure 17b).

The presence of silicone outside the implant may induce an inflammatory reaction, leading to the formation of a silicone granuloma that can present contrast enhancement, thus causing diagnostic problems (Figure 18) [90,91]. Not infrequently, percutaneous biopsy is warranted to differentiate between silicone granuloma and local relapse.

Although MRI has a high negative predictive value of 98%, it has a low positive predictive value of 77% [92] and one should be aware that there are pitfalls in image interpretation, like overestimation of contour abnormalities or long and complex radial folds, and the diagnosis of rupture should not be supported by only a single imaging finding (Table 3).

#### 5.2.6. Breast Implant Associated Atypical Large Cell Lymphoma (BIA-ALCL)

BIA-ALCL was first described in 1997 [93] but it was recognized in 2016 by the World Health Organization (WHO) as a unique T-cell ALK-negative ALCL, with similar morphologic and immunophenotypic characteristics to systemic and cutaneous ALK-negative BIA-ALCL [94]. The estimated incidence is one to three cases per million women and it presents, on average, 11 years after implantation, with higher incidence reported in textured implants [95,96].

The pathogenesis is not fully elucidated, yet two hypotheses have been described. The first presents the Gram-negative Ralstonia bacterium as a causative factor, which is frequently encountered adjacent to the textured breast implants in patients with BIA-ALCL, while the second theory implicates silicone bleed and leakage of microparticles as the trigger factor. In both cases, chronic inflammation is induced with repetitive T-cell activation [97].

The most common clinical presentations are unilateral breast edema and, occasionally, a palpable mass adjacent to the implant. Two different types of BIA-ALCL are described: peri-implant effusion and peri-implant mass. MRI and US are the most accurate imaging modalities for the detection of effusion and mass (US 84% and 46%, MRI 82% and 50%, respectively). On US, the most common findings are a homogeneous peri-implant effusion and, occasionally, a solid oval mass, with well-defined margins [98].

MRI is the second in-line modality for BIA-ALCL diagnosis with the standard protocol before and after contrast administration. The main findings are the presence of fluid (with hyperintensity in T2 images, homogeneous or heterogeneous) between the capsule and the implant, and peri-implant masses, round or irregular with heterogeneous enhancement. Commonly, the capsule presents with irregular thickness and contrast enhancement (Figure 19) [99].

BIA-ALCL is a rare entity that radiologists should suspect in the case of delayed peri-implant effusion, and cytology or tissue sampling should be performed in the case of a suspicious peri-implant mass.

#### 5.2.7. Others

Breast implant associated squamous cell carcinoma (BIA-SCC) is a very rare but potentially aggressive epithelial tumor emanating from the breast implant capsule, with sheets of squamous cells varying from normal to dysplasia, metaplasia, and carcinoma. The clinical presentation of BIA-SCC is very similar to BIA-ALCL and BIA-SCC should be also considered in the case of late onset seroma and US and MRI should be performed [100].

Desmoid tumors of the breast are extremely rare, accounting for 0.2% of all breast tumors and 4% of all extra-abdominal desmoid tumors. The presence of silicone prostheses is reported as a risk factor for developing breast desmoid tumors arising from the fibrous capsule as a result of postsurgical trauma [101]. They are best evaluated in MRI and have two distinct features: chest wall tumors presenting as oval and lobulated masses, which are locally aggressive invading the intercostal muscles and pleura, and breast tumors presenting as spiculated masses. The masses are isointense to muscle in T1 images with variable T2 signal intensity and heterogeneous enhancement [102].

## 6. Recurrence of Breast Cancer

Mastectomy significantly decreases the risk for breast cancer, but it does not eliminate it, since residual glandular tissue can remain even at the microscopic level. Recurrence rates after mastectomy, with or without reconstruction, are 1–2% annually for the first 5 years, and the overall recurrence rate is 2–15% [103]. The incidence of local recurrence after mastectomy without reconstruction is higher than in women with breast reconstruction (2–7.5% vs. 2–4%), with comparable rates for delayed and immediate reconstruction [104,105]. Reported risk factors for local recurrence are whether the patient is aged younger than 50 years old at the time of diagnosis, large tumor size, and aggressive molecular subtypes [104,106].

The recurrences are commonly located superficially in the skin and the subcutaneous tissue (60%) and are easily detected with clinical examination, while in 32.5% of cases the location is deep adjacent to the pectoralis muscle and it can be occult, masked by the autologous tissue or implant [104]. The European Society for Radiotherapy and Oncology (ESTRO) recently issued guidelines for the clinical target volume for postmastectomy radiation therapy after immediate implant reconstruction that encompasses the location of most local recurrences [107,108].

The clinical benefit of imaging surveillance of women with breast reconstruction is under intense debate with conflicting suggestions in the literature. The National Comprehensive Cancer Network (NCCN) advises against imaging for asymptomatic patients with breast cancer treated with mastectomy with or without reconstruction, while the American College of Radiology (ACR) recommends surveillance with mammography or digital tomosynthesis for women with mastectomy and autologous reconstruction with or without prosthesis [103]. On the other hand, Adrada et al. [104] and Pinel-Giroux et al. [51] support the use of MRI in women with high recurrence risk. Although the pooled overall cancer detection rate per 1000 examinations of MRI is higher than mammography and US (5.17, 1.86 and 2.66 respectively) [109], the level of evidence in the literature is too low to dictate a surveillance strategy in this context, and further studies are required with a special focus on prognosis and cost-effectiveness.

The imaging appearance of tumor recurrence is similar to breast cancer as an irregular mass on all imaging modalities, but radiologists should be aware that in 50% of cases, the mass has a pseudo-benign appearance [104]. US is used as the first-line method for evaluation of palpable masses, and the majority of lesions are hypoechoic with 8.6% being complex cystic masses [110]. MRI is the most sensitive method to detect tumor recurrence, irrespective of tumor location (deep or superficial) and type of reconstruction (implant or autologous) (Figure 20) [104].

## 7. Male Breast

Breast cancer is rare in men, accounting for less than 1% of all breast cancers. Due to the typical central retro-areolar location and the frequent involvement of the nipple, the standard surgical treatment of male breast cancer is a modified radical mastectomy with excision of the nipple and axillary node dissection [111]. Although scarce, there are reports in the literature regarding postmastectomy breast reconstruction in men. Autologous reconstruction is the most common method, with local flaps followed by the TRAM flap and there is a single report of fat grafting for chest symmetry [112,113].

## 8. Conclusions

Breast reconstruction based on either autologous or alloplastic techniques has evolved tremendously in recent years, offering a variety of options tailored to the specific needs of each patient. Plastic surgery in this field aims to provide an improved aesthetic outcome while minimizing postoperative complications, always within the context of oncological safety. Although there are no consensus and guidelines regarding the imaging surveillance of post-mastectomy patients that underwent reconstructive surgery, it is important to be familiar with the normal imaging appearance of different reconstructive techniques and to be able to recognize associated complications, either benign or malignant. One must always keep in mind that even though the risk of recurrent breast cancer is radically minimized after mastectomy, it is not eliminated and can be clinically occult.

## Figures and Tables

**Figure 1 diagnostics-13-03186-f001:**
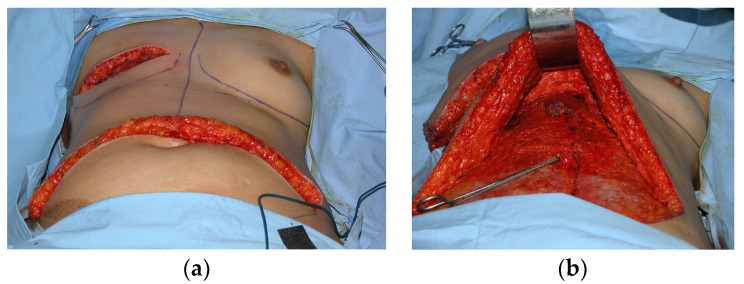
Intraoperative images of transverse abdominal myocutaneous (TRAM) flap reconstruction after right mastectomy (**a**) before and (**b**) after transfer to the breast.

**Figure 2 diagnostics-13-03186-f002:**
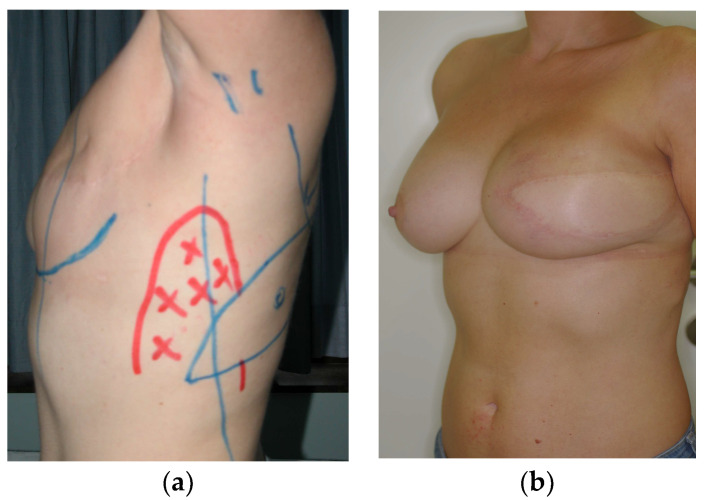
(**a**) Preoperative design with skin paddle and area of no dissection and (**b**) postoperative image of left latissimus dorsi (LD) flap combined with an implant for the reconstruction of the lest breast.

**Figure 3 diagnostics-13-03186-f003:**
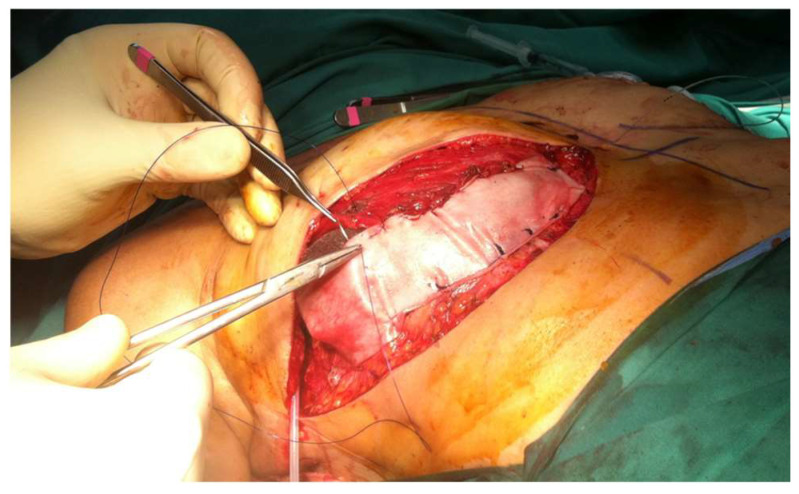
Dual plane use of acellular dermal martice (ADM) to support and cover the lower pole of the reconstruction between the pectoralis major muscle edge and the inframammary fold.

**Figure 4 diagnostics-13-03186-f004:**
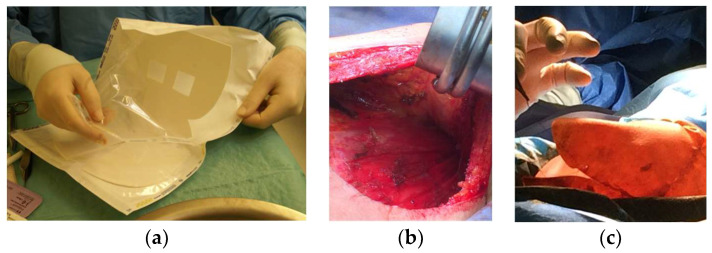
(**a**) ADM for circumferential coverage (Braxon, Decomed), (**b**) prepectoral mastectomy cavity to reconstruct, (**c**) preparation of ravioli to be inserted in the mastectomy cavity to be reconstructed.

**Figure 5 diagnostics-13-03186-f005:**
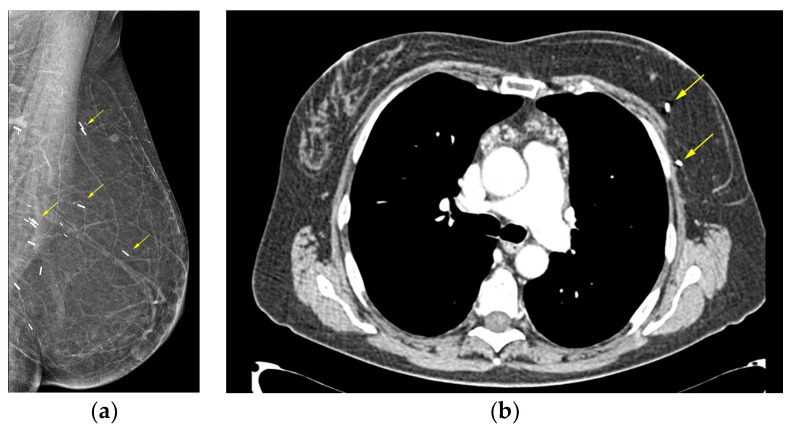
Deep inferior epigastric perforator (DIEP) reconstruction: (**a**) mediolateral oblique mammographic (MLO) view of the reconstructed breast depicts essentially fatty tissue and surgical clips (arrows); (**b**) axial thoracic CT scan confirms the presence of a viable DIEP reconstruction with no complications.

**Figure 6 diagnostics-13-03186-f006:**
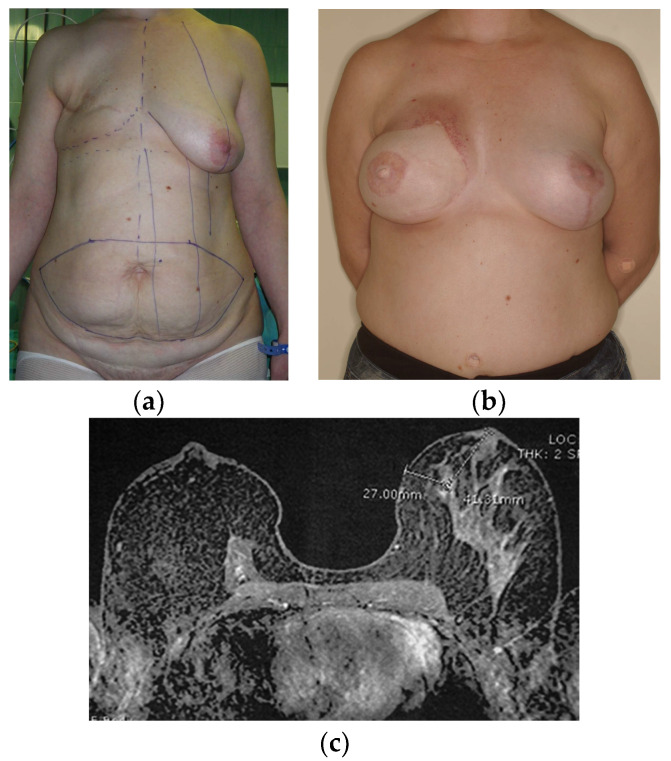
(**a**) Preoperative design and (**b**) postoperative image of right mastectomy defect reconstruction with a pedicled TRAM flap and left breast mastopexy. (**c**) On MRI, the right reconstructed breast consists primarily of fat and the muscle is recognized centrally anteriorly to the chest wall.

**Figure 7 diagnostics-13-03186-f007:**
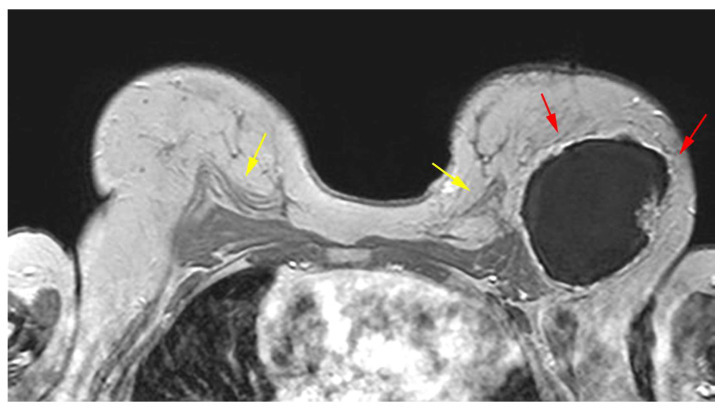
Bilateral autologous LD reconstruction. Yellow arrows point to the characteristic tailed aspect of the muscle. Red arrows depict a hypointense circumscribed area with irregular wall enhancement and associated solid-enhancing nodule corresponding to a local relapse of high grade invasive ductal carcinoma, triple negative, with marked central necrosis.

**Figure 8 diagnostics-13-03186-f008:**
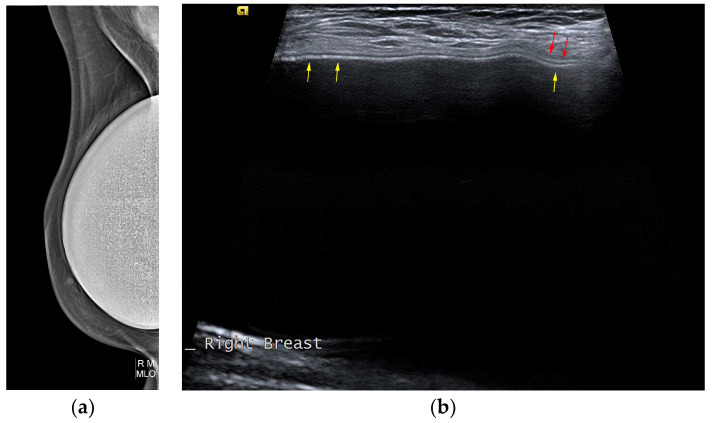
Mediolateral oblique MLO mammography view and ultrasound (US) of the right breast after mastectomy and reconstruction with a single-lumen silicone implant: (**a**) in the MLO view, the implant appears as a homogenous radiopaque oval mass with smooth borders; (**b**) on US the implant is anechoic. The intact shell appears as an echogenic line (yellow arrows) and the fibrous capsule is seen as a parallel echogenic line (red arrows).

**Figure 9 diagnostics-13-03186-f009:**
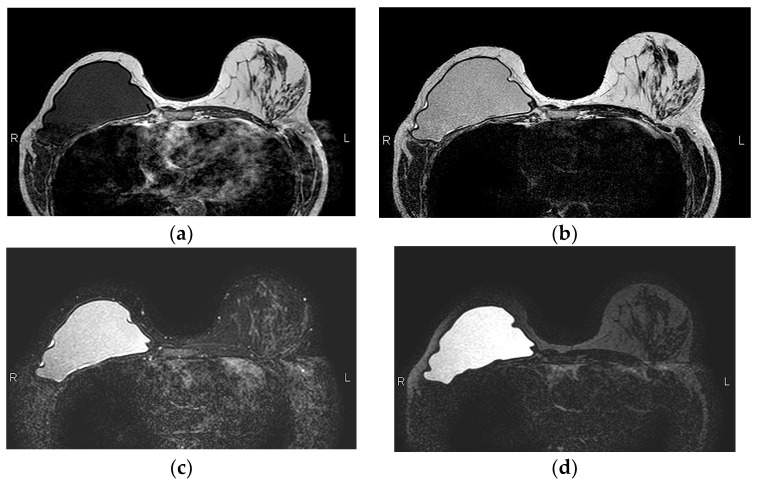
Normal MRI appearance of an intact single-lumen silicone implant of the reconstructed right breast: (**a**) in the T1 non-fat-saturated sequence, the silicone has low signal intensity; on T2-weighted images, both non-fat-saturated (**b**) and fat-saturated (**c**), the silicone has high signal intensity. Minimal peri-implant fluid is a common normal finding with high signal intensity; (**d**) in the silicone-selective sequence, the silicone appears with high signal intensity while the peri-implant fluid has low signal intensity.

**Figure 10 diagnostics-13-03186-f010:**
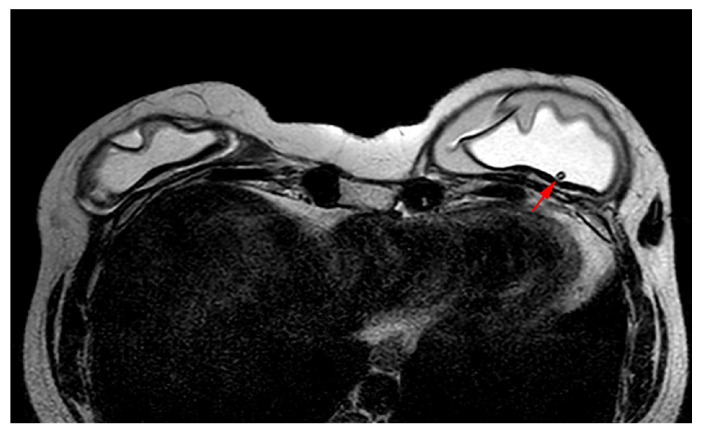
MRI appearance of double-lumen implant (silicone in the outer lumen, saline in the inner lumen). The red arrow points at the valve used to inflate the inner saline part.

**Figure 11 diagnostics-13-03186-f011:**
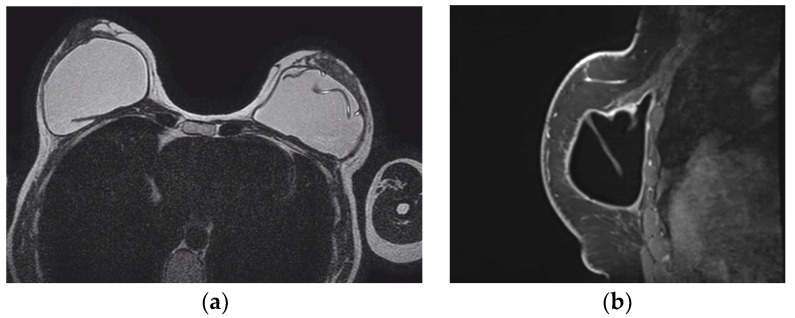
MRI appearance of single-lumen saline implants with radial fold of the right implant and intracapsular rupture of the left implant: (**a**) axial T2 sequence depicts a line perpendicular to the implant shell line, corresponding to a radial fold, whereas the curvilinear line, parallel to the shell, indicates a subcapsular line, in concordance with an intracapsular rupture; (**b**) sagittal view confirms the “perpendicular to implant-shell” character of the radial fold of the right implant. Sagittal views can be of help in differentiating radial folds from true intracapsular rupture.

**Figure 12 diagnostics-13-03186-f012:**
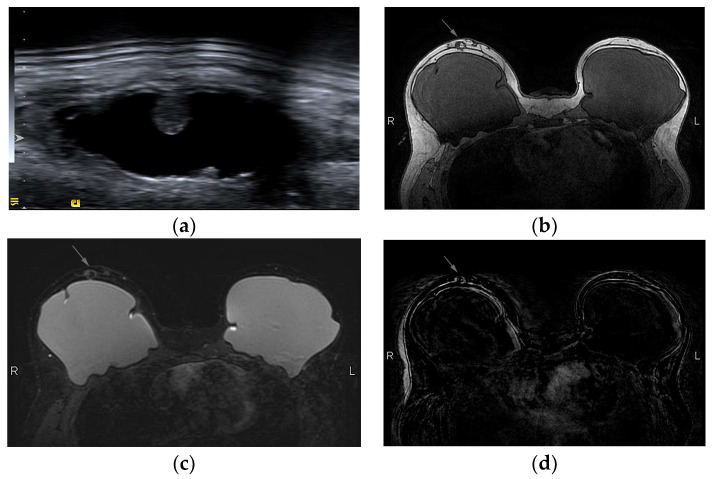
(**a**) Fat necrosis in the right breast following bilateral mastectomy and implant reconstruction depicted as a complex cystic mass on US. (**b**) On T1-weighted images, the mass is isointense to fat and (**c**) has markedly low signal intensity on STIR sequence, known as the “black hole” sign. (**d**) On subtraction MRI the mass exhibits no enhancement.

**Figure 13 diagnostics-13-03186-f013:**
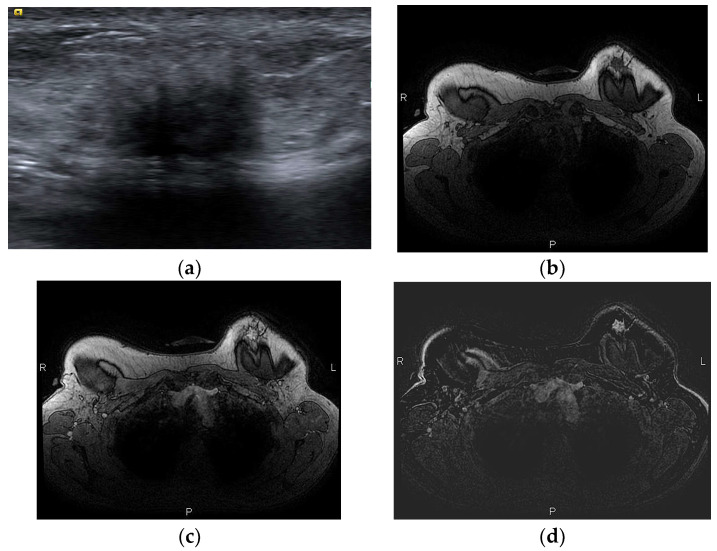
A woman with a history of left breast cancer presented with a palpable mass 2 years after bilateral mastectomy and implant reconstruction: (**a**) on US, the mass was hypoechoic with indistinct and irregular margins; (**b**) on T1-weighted MR images, the mass is hypointense with an irregular shape and spiculated margins; (**c**) on DCE MRI, 3 min after contrast administration and (**d**) subtraction image, the mass is strongly and homogeneously enhancing. The diagnosis was fat necrosis after core biopsy.

**Figure 14 diagnostics-13-03186-f014:**
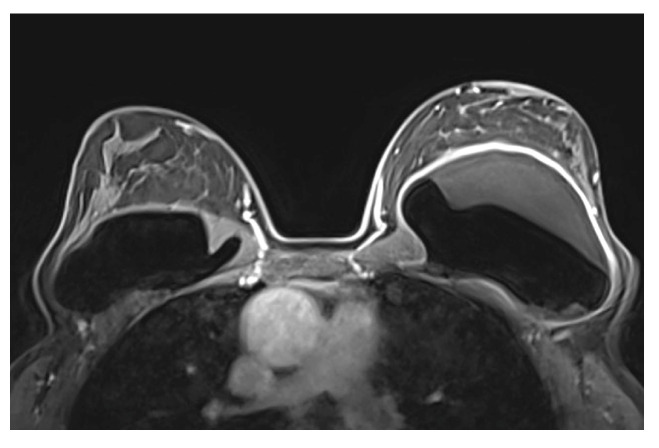
MRI appearance of left implant infection. Axial T1-weighted fat-saturation post-contrast view of a left peri-implant inflammatory infusion with peripheral smooth contrast uptake.

**Figure 15 diagnostics-13-03186-f015:**
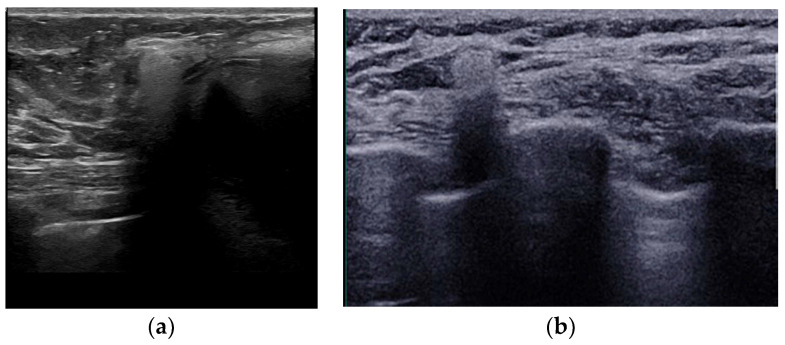
Ultrasound appearance of extracapsular rupture. (**a**) Free silicone in the breast along the upper outer part of the implant, indicating extracapsular rupture. Typical appearance of a marked hyperechoic area with posterior shadowing, referred to as “snowstorm appearance”. (**b**) “Snowstorm” appearance of an axillary lymph node in another patient. This was an isolated finding, and no rupture was documented on MRI. This was related to silicone gel bleed, i.e., microscopic diffusion of silicone molecules through the semipermeable weakened but intact implant elastomer shell.

**Figure 16 diagnostics-13-03186-f016:**
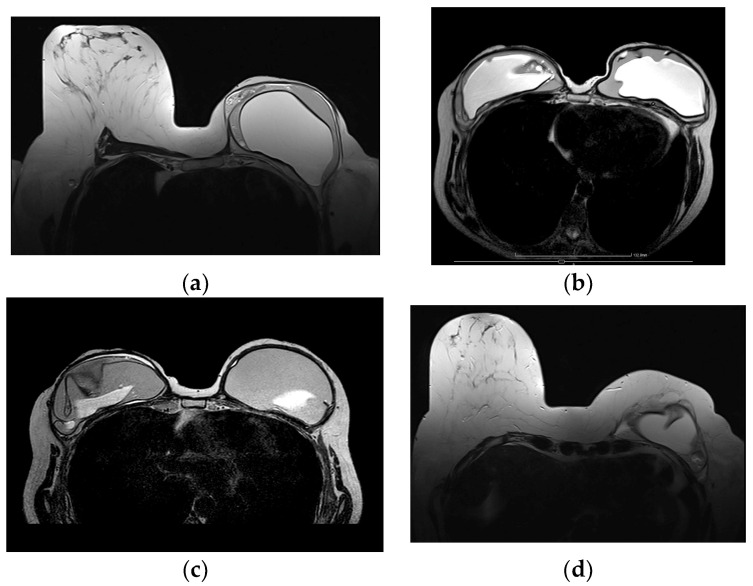
MRI appearance of various signs indicative of intracapsular and extracapsular rupture. All images are axial T2-weigthed images from different patients. (**a**) Salad oil and subcapsular line sign, evocating intracapsular rupture. (**b**) Isolated salad-oil sign in both implants. This sign is not sufficient per se to validate an intracapsular rupture. (**c**) Key-hole sign, lasso sign and linguni sign, indicating intracapsular rupture of the right implant. (**d**) Extracapsular rupture can be safely indicated on this characteristic aspect of extra-implant presence of silicone.

**Figure 17 diagnostics-13-03186-f017:**
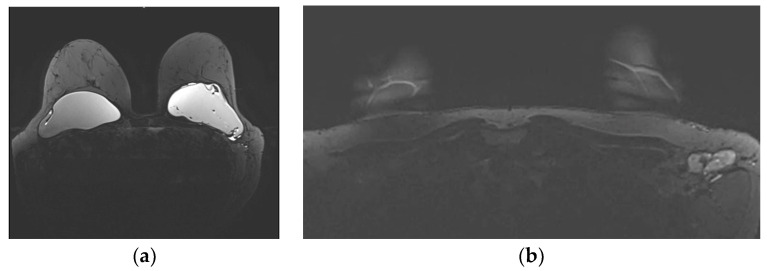
MRI appearance of extracapsular rupture. (**a**) Axial silicone-only sequence depicts hyperintense free droplets of silicone outside the left implant capsule. (**b**) Same patient, axial silicone-only sequence through the axillary level depicts marked hyperintense ipsilateral axillary lymph nodes, evocating silicone deposits.

**Figure 18 diagnostics-13-03186-f018:**
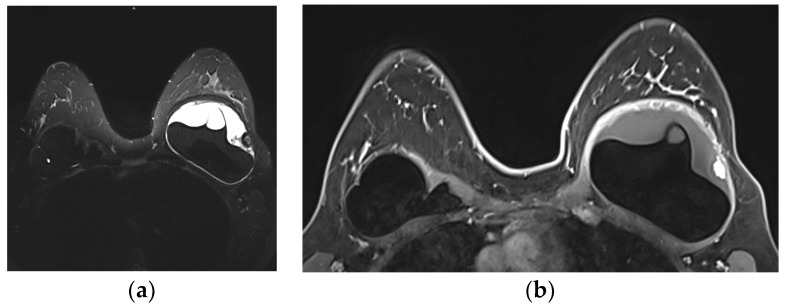
Silicone granuloma associated with peri-implant effusion and mild inflammation. (**a**) Axial short tau inversion recovery (STIR) sequence depicts hyperintense peri-implant effusion associated with a hypointense nodular component. (**b**) Axial T1 fat-saturated post-contrast sequence readily depicts the strong enhancement of the nodular component and the mild enhancement of the peri-implant effusion collection. This aspect could evocate a silicone granuloma; nevertheless, a core biopsy was performed to validate this diagnosis.

**Figure 19 diagnostics-13-03186-f019:**
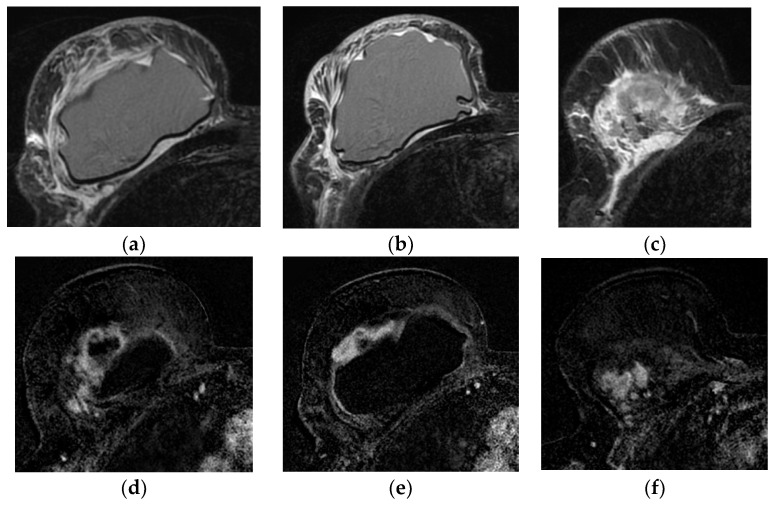
Breast implant associated atypical large cell lymphoma (BIA-ALCL), presented 10 years after implantation as a peri-implant mass, associated with pronounced edema of the breast and pectoralis major muscle in T2 MRI images (**a**–**c**). (**d**–**f**) In subtraction images, the mass is irregular with strong contrast enhancement and is invading the pectoral muscle (**f**). Images are courtesy of Prof. M. Fuchsjäger, Graz, Austria.

**Figure 20 diagnostics-13-03186-f020:**
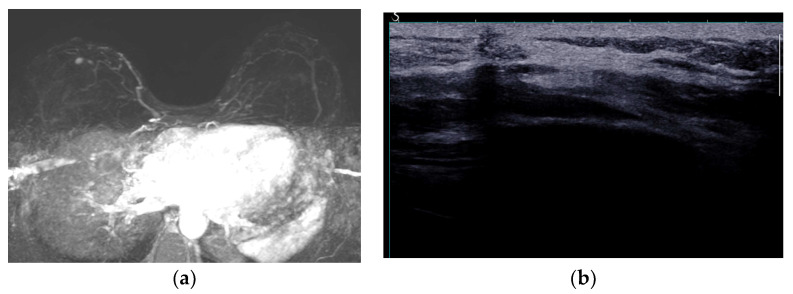
BRCA1 carrier. Right mastectomy due to extensive comedo-DCIS and autologous reconstruction. Prophylactic left mastectomy and reconstruction. (**a**) Maximum intensity projection (MIP) images depict a small enhancing nodule, with a millimetric satellite lesion, both of which are suspicious in this context. (**b**) Second-look post MRI ultrasound depicts a hypoechoic, taller-than-wider solid nodule with irregular margins. Core biopsy confirmed local relapse of high-grade DCIS.

**Table 1 diagnostics-13-03186-t001:** Standard MRI protocol for implant-based breast reconstruction.

Sequence	T1 TSE	T2 TSE	Dynamic 3D Fat Suppressed	DWI	Silicone Suppression (Fat Suppressed)	Silicone Excited (Water Suppressed)
Slice thickness (mm)	2	4	2	4	4	4
FOV (mm)	360	360	220	360	360	360
TR	5.9	3500	4.8	3520	6470	4500
TE	2.46	86	1.73	54	88	54
Time (min)	1:19	2:27	6:46	4:00	4:46	3:20

FOV—Field Of View, TSE—turbo spin echo, DWI—diffusion weighted imaging.

**Table 2 diagnostics-13-03186-t002:** Diagnostic checklist for breast MRI after implant reconstruction.

Checklist	Findings	Details	Diagnosis
Implant type	One compartment Two compartments		Single-lumen Double-lumen
Implant composition	Saline	High T2; low T1	
	Silicone	High T2; hyperintense silicone-only sequences; hypointense silicone-suppression sequences	
Implant location	Pre-pectoral Retro-pectoral		
Normal findings	Thin fibrous capsule Small peri-implant fluid		
	Radial folds		
Rupture *	Implant shell	Intact fibrous capsule	Intracapsular
	Implant shell and fibrous capsule		Extracapsular
Complications	Fluid	Early; low T1, high T2	Seroma
		Early; high T1, high T2	Hematoma
		Delayed, persistent (±mass)	BIA-ALCL, BIA-SCC
	Thickened capsule	Irregular	Capsular contracture
		Enhancing with fluid	Infection
	Mass	Cyst, complex cystic	Fat necrosis
		Fibrous capsule related	BIA-ALCL, BIA-SCC, desmoid, granuloma
		Skin, subcutaneous	Fat necrosis, cancer recurrence, granuloma

* Details on implant rupture signs are shown in Table 3. BIA-ALC: breast implant associated atypical large cell lymphoma; BIA-SCC: breast implant associated squamous cell carcinoma.

**Table 3 diagnostics-13-03186-t003:** MRI signs of implant rupture.

	Signs	Details
Definitive rupture	Linguini sign	Low signal lines within the silicone
	Subcapsular lines	Low signal lines parallel to the capsule surrounded by silicone
	Free Silicone	Extracapsular silicone
Possible rupture	Keyhole or teardrop Sign	Focal silicone invagination between implant shell and fibrous capsule
	Salad-oil sign Rat-tail sign	Silicone gel mixed with peri-implant fluid Contour irregularity extending along the chest wall
Pitfalls	Complex radial folds	Long invaginations of the shell
	Gel bleed	Silicone within the axillary lymph nodes
	Double-lumen implant	Inner compartment shell may mimic intracapsular rupture

## Data Availability

Not applicable.

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
