# Peer review of "Imaging of the Reconstructed Breast"

_diagnostics, 2023, doi:10.3390/diagnostics13203186_

Round 1
Reviewer 1 Report
This is quite an extensive and content-rich review, aiming to share insights regarding post-mastectomy breast reconstruction imaging. The author presents a variety of MRI, ultrasound, and mammography images in an effort to contribute knowledge to this field.
However, for a topic such as "Imaging of Reconstructed Breast" that aims to be published in 2023, it may require a higher degree of specificity to offer readers enhanced value. The primary reason is that this domain already boasts numerous commendable articles, some published two years ago and even others dating back twelve years. If we intend to write on such a topic today, there must be unique highlights that surpass previous literature and reflect the state of affairs in 2023.
To achieve stronger specificity, examples could include the author's mention of the importance of understanding implants (line 300), yet the author does not provide a detailed introduction to the various types of breast implants. Particularly, the rise in popularity of newer implant types like Motiva in recent years is something that an article attempting to teach "Imaging of Reconstructed Breast" in 2023 should acknowledge.
Alternatively, the author could offer a comprehensive diagnostic imaging protocol, such as MRI scan parameter settings, key interpretation details, or even a checklist to assist newcomers entering this field.
Furthermore, the author could provide surgical illustrations to help imaging-diagnostic physicians understand how the surgical process that we observe in static medical images was conducted in the past and how it might develop in the future.
As it stands, the current work appears more like a compilation of various sporadic images the author has encountered, supplemented by extensive text incorporating surgical background knowledge and points the author considers vital. However, it lacks a stronger educational significance or diagnostic framework beyond past literature.
Especially, the extensive introduction of surgical knowledge is primarily textual and not strongly correlated with the various diagnostic images provided later on. Not all the complications the author mentions have associated images for educational purposes. These are areas that can be improved upon.
While it is understandable that the author dedicated a considerable amount of time to gather these images, for a review article, offering a stronger educational intent or more profound diagnostic insights would better justify the necessity for publication.
- Adrada BE, Whitman GJ, Crosby MA, Carkaci S, Dryden MJ, Dogan BE. Multimodality Imaging of the Reconstructed Breast. Curr Probl Diagn Radiol. 2015 Nov-Dec;44(6):487-95. doi: 10.1067/j.cpradiol.2015.04.006. Epub 2015 Apr 27. PMID: 26118619; PMCID: PMC4567950.
- Krisnan RNK, Chotai N. Imaging Spectrum of Augmented Breast and Post-Mastectomy Reconstructed Breast with Common Complications: A Pictorial Essay. Korean J Radiol. 2021 Jul;22(7):1005-1020. doi: 10.3348/kjr.2020.0779. Epub 2021 Apr 9. PMID: 33938642; PMCID: PMC8236364.
- Dialani V, Lai KC, Slanetz PJ. MR imaging of the reconstructed breast: What the radiologist needs to know. Insights Imaging. 2012 Jun;3(3):201-13. doi: 10.1007/s13244-012-0150-7. Epub 2012 Mar 17. PMID: 22696083; PMCID: PMC3369124.
- Scaranelo AM, Lord B, Eiada R, Hofer SO. Imaging approaches and findings in the reconstructed breast: a pictorial essay. Can Assoc Radiol J. 2011 Feb;62(1):60-72. doi: 10.1016/j.carj.2010.09.010. Epub 2010 Nov 10. PMID: 21067890.
- Adrada BE, Karbasian N, Huang M, Rauch GM, Woodtichartpreecha P, Whitman G. Imaging Surveillance of the Reconstructed Breast in a Subset of Patients May Aid in Early Detection of Breast Cancer Recurrence. J Clin Imaging Sci. 2021 Nov 9;11:58. doi: 10.25259/JCIS_113_2021. PMID: 34877066; PMCID: PMC8645461.
- Yoo H, Kim BH, Kim HH, Cha JH, Shin HJ, Lee TJ. Local recurrence of breast cancer in reconstructed breasts using TRAM flap after skin-sparing mastectomy: clinical and imaging features. Eur Radiol. 2014 Sep;24(9):2220-6. doi: 10.1007/s00330-014-3214-x. Epub 2014 May 24. PMID: 24852813.
Author Response
Reviewer 1:
This is quite an extensive and content-rich review, aiming to share insights regarding post-mastectomy breast reconstruction imaging. The author presents a variety of MRI, ultrasound, and mammography images in an effort to contribute knowledge to this field.
However, for a topic such as "Imaging of Reconstructed Breast" that aims to be published in 2023, it may require a higher degree of specificity to offer readers enhanced value. The primary reason is that this domain already boasts numerous commendable articles, some published two years ago and even others dating back twelve years. If we intend to write on such a topic today, there must be unique highlights that surpass previous literature and reflect the state of affairs in 2023.
To achieve stronger specificity, examples could include the author's mention of the importance of understanding implants (line 300), yet the author does not provide a detailed introduction to the various types of breast implants. Particularly, the rise in popularity of newer implant types like Motiva in recent years is something that an article attempting to teach "Imaging of Reconstructed Breast" in 2023 should acknowledge.
Response: We would like to thank you for the comment. A more detailed and updated presentation of the types of implants is added in page 9, lines 334-357
Alternatively, the author could offer a comprehensive diagnostic imaging protocol, such as MRI scan parameter settings, key interpretation details, or even a checklist to assist newcomers entering this field.
Response: A proposed MRI protocol is presented in Table 1. Table 2 presents a diagnostic checklist for breast MRI after implant reconstruction with key interpretation details, as suggested by the reviewer, and table 3 summarizes the MRI signs of implant rupture.
Furthermore, the author could provide surgical illustrations to help imaging-diagnostic physicians understand how the surgical process that we observe in static medical images was conducted in the past and how it might develop in the future.
Response: Images at figures 1, 2, 3 and 4 are presenting the surgical procedures regarding the autologous and alloplastic breast reconstruction.
As it stands, the current work appears more like a compilation of various sporadic images the author has encountered, supplemented by extensive text incorporating surgical background knowledge and points the author considers vital. However, it lacks a stronger educational significance or diagnostic framework beyond past literature.
Especially, the extensive introduction of surgical knowledge is primarily textual and not strongly correlated with the various diagnostic images provided later on. Not all the complications the author mentions have associated images for educational purposes. These are areas that can be improved upon.
Response: Thank you very much for these comments. More images are added as aforementioned presenting the surgical procedures that are not available at previous published articles, and we hope that they enhance the educational purpose of this review article. Moreover, as suggested by the reviewer we added figures 12, 13 and 19 enriching the images of complications.
While it is understandable that the author dedicated a considerable amount of time to gather these images, for a review article, offering a stronger educational intent or more profound diagnostic insights would better justify the necessity for publication.
- Adrada BE, Whitman GJ, Crosby MA, Carkaci S, Dryden MJ, Dogan BE. Multimodality Imaging of the Reconstructed Breast. Curr Probl Diagn Radiol. 2015 Nov-Dec;44(6):487-95. doi: 10.1067/j.cpradiol.2015.04.006. Epub 2015 Apr 27. PMID: 26118619; PMCID: PMC4567950.
Krisnan RNK, Chotai N. Imaging Spectrum of Augmented Breast and Post-Mastectomy Reconstructed Breast with Common Complications: A Pictorial Essay. Korean J Radiol. 2021 Jul;22(7):1005-1020. doi: 10.3348/kjr.2020.0779. Epub 2021 Apr 9. PMID: 33938642; PMCID: PMC8236364.
- Dialani V, Lai KC, Slanetz PJ. MR imaging of the reconstructed breast: What the radiologist needs to know. Insights Imaging. 2012 Jun;3(3):201-13. doi: 10.1007/s13244-012-0150-7. Epub 2012 Mar 17. PMID: 22696083; PMCID: PMC3369124.
- Scaranelo AM, Lord B, Eiada R, Hofer SO. Imaging approaches and findings in the reconstructed breast: a pictorial essay. Can Assoc Radiol J. 2011 Feb;62(1):60-72. doi: 10.1016/j.carj.2010.09.010. Epub 2010 Nov 10. PMID: 21067890.
- Adrada BE, Karbasian N, Huang M, Rauch GM, Woodtichartpreecha P, Whitman G. Imaging Surveillance of the Reconstructed Breast in a Subset of Patients May Aid in Early Detection of Breast Cancer Recurrence. J Clin Imaging Sci. 2021 Nov 9;11:58. doi: 10.25259/JCIS_113_2021. PMID: 34877066; PMCID: PMC8645461.
- Yoo H, Kim BH, Kim HH, Cha JH, Shin HJ, Lee TJ. Local recurrence of breast cancer in reconstructed breasts using TRAM flap after skin-sparing mastectomy: clinical and imaging features. Eur Radiol. 2014 Sep;24(9):2220-6. doi: 10.1007/s00330-014-3214-x. Epub 2014 May 24. PMID: 24852813.
Response: Thank you for the comment. The relevant articles have been added.
Reviewer 2 Report
This is an interesting, well-written review aiming to present the various types of breast reconstruction with emphasis on pertinent imaging findings and complications. The review and discussion are nicely presented and I feel that it is an important and interesting paper to the field. But I have some suggestions and minor comments to be completed before review is accepted:
- I would advise the authors to highlight and emphasize the advantages of your review over other publicly available papers.
- Is it possible to use only MRI and ultrasound to reconstructed breast imaging? It is worth briefly describing other possible experimental or clinical technologies.
- Line 223 and 231. Section numbers need to be corrected.
- Line 260 ‘Postoperative scarring and clips are common findings’. It is necessary to indicate the possible impact of clips on complications.
- The Complications section describes the MRI signs of skin thickening / fibrosis / fat necrosis very well, but images are lacking. I would suggest adding them.
Author Response
Reviewer 2:
This is an interesting, well-written review aiming to present the various types of breast reconstruction with emphasis on pertinent imaging findings and complications. The review and discussion are nicely presented and I feel that it is an important and interesting paper to the field. But I have some suggestions and minor comments to be completed before review is accepted:
- I would advise the authors to highlight and emphasize the advantages of your review over other publicly available papers.
Response: You would like to thank you for your comment. This review article presents the imaging findings of the reconstructed breast and in addition it emphasizes on the presentation of the surgical procedures, that are not quite extensively presented on previous published articles.
- Is it possible to use only MRI and ultrasound to reconstructed breast? It is worth briefly describing other possible experimental or clinical technologies.
Response: As it is discussed in the manuscript the clinical benefit of imaging surveillance of with breast reconstruction is under intense debate with conflicting suggestions at the without definite guidelines. The role of mammography is quite limited and US can be used to evaluate the reconstructed breast, but MRI is considered the method of choice. To our knowledge there are no reported experimental technologies on this field/
- Line 223 and 231. Section numbers need to be corrected.
Response: Thank you very much, the section number is corrected.
- Line 260 ‘Postoperative scarring and clips are common findings’. It is necessary to indicate the possible impact of clips on complications.
Response: The presence of postoperative clips is simply recorded, there is no correlation with complications neither a potential pitfall.
- The Complications section describes the MRI signs of skin thickening / fibrosis / fat necrosis very well, but images are lacking. I would suggest adding them.
Response: Thank you for this comment. Figures 12 and 13 have been added.
Reviewer 3 Report
In the manuscript entitled “Imaging of the Reconstructed Breast,” the authors offer an abundant search of information on the imaging advances of reconstructed breasts.
Even with the strong amount of information and the organization of the manuscript, the following points must be addressed before the manuscript is accepted for publication in Diagnostics MDPI.
1. The title must be more specific to bring insight to the reader about the content of the review. For example “MR imaging…” thus, the reader can expect the kind of technique for analyzing the image referred to in the text. In addition, the authors claim “The identification of breast cancer recurrence and malignant disease associated with breast implants is another issue that is discussed in this review.” Thus, the title needs to reflect that content.
2. Since there are similar works already published −p.e. Dialani V, Lai KC, Slanetz PJ. MR imaging of the reconstructed breast: What the radiologist needs to know. Insights Imaging, 3-3(2012), pp. 201-13. − The authors need to highlight the novelty of the work.
3. I recommend adding a section devoted to the discussion of emerging techniques for the imaging of the reconstructed breast.
4. I recommend adding a section devoted to the mistakes made in the instrument for imaging the reconstructed breast. This section would be useful for the reader, particularly for researchers focused on artificial intelligence systems for the automatic interpretation of images.
5. Section 7 “Male breast” must be enhanced, it would be desirable to add some cases and the corresponding images.

none
Author Response
Reviewer 3:
In the manuscript entitled “Imaging of the Reconstructed Breast,” the authors offer an abundant search of information on the imaging advances of reconstructed breasts.
Even with the strong amount of information and the organization of the manuscript, the following points must be addressed before the manuscript is accepted for publication in Diagnostics MDPI.
- The title must be more specific to bring insight to the reader about the content of the review. For example “MR imaging…” thus, the reader can expect the kind of technique for analyzing the image referred to in the text. In addition, the authors claim “The identification of breast cancer recurrence and malignant disease associated with breast implants is another issue that is discussed in this review.” Thus, the title needs to reflect that content.
Response: You would like to thank the reviewer for this comment. This title mirrors the main topic of this review article, the imaging findings of the reconstructed breast with all the imaging modalities. An overview of the surgical procedures is presented that aims to familiarize the reader with the different techniques that are currently used, in order to recognize the normal appearance of the reconstructed breast and their common complications. In our opinion it is not possible to encompass the content of this article in one title.
- Since there are similar works already published −p.e. Dialani V, Lai KC, Slanetz PJ. MR imaging of the reconstructed breast: What the radiologist needs to know. Insights Imaging, 3-3(2012), pp. 201-13. − The authors need to highlight the novelty of the work.
- I recommend adding a section devoted to the discussion of emerging techniques for the imaging of the reconstructed breast.
Response to comments 2 and 3: The past few years there is no dramatic evolution on breast reconstruction. The latest advances on the surgical procedures are highlighted as well as the current status on imaging surveillance.
- I recommend adding a section devoted to the mistakes made in the instrument for imaging the reconstructed breast. This section would be useful for the reader, particularly for researchers focused on artificial intelligence systems for the automatic interpretation of images.
Response: Thank you for the comment. In our knowledge there are no published studies on AI on reconstructed breasts. There can be pitfalls in the interpretation of implant rupture, presented in table 3.
- Section 7 “Male breast” must be enhanced, it would be desirable to add some cases and the corresponding images.
Response: Only a few cases of male breast reconstruction are reported in the literature and unfortunately, we do not have any clinical practice on male breast reconstruction.
Round 2
Reviewer 1 Report
The author's revision has incorporated numerous surgical images and added several new paragraphs, demonstrating a dedicated effort to enhance the quality of this article. The final outcome indeed shows significant improvement. However, from my perspective, there are still some minor issues with verbosity and fragmentation.
As for whether the current version meets the requirements of the journal, I leave that decision to the academic editor.
The formatting of the references section appears somewhat loose, possibly due to alignment issues. I believe that if accepted, the journal's copyeditor should be able to address this effectively.
Generally okay. Minor editing of English language might be required